# Financial and Economic Assessment of Tidal Stream Energy—A Case Study

Stocker Klaus

Department of International Business, Georg-Simon Ohm Institute of Technology, D-90489 Nuremberg, Germany; klaus.stocker@th-nuernberg.de

**Abstract:** This case study is based on actual project and consultancy work, balancing real life experience with a review and analysis of empirical and theoretical literature. Tidal stream energy (TSE) is still a nascent technology, but with much better predictability than the classical alternatives of sun and wind. Being still more expensive than other renewable technologies, it is important to find locations in order to initiate a learning process to bring down cost to a competitive level as it was the case for solar and wind technologies. Locations for an initial phase of operation of TSE small islands in the Philippines (and other Asian countries) were found to be most suitable, because expensive and polluting diesel generators can be replaced and a reliable 24 h electricity supply can be established. Different appraisal methods in different scenarios show that under normal circumstances a hybrid combination of TSE, solar energy and battery storage is financially and economically superior to existing fossil energy based power stations as well as to solar energy alone. However, the traditional financial approaches are not always reliable, in spite of superficial mathematical exactness, and the parameters used must be analysed carefully, especially if we deal with innovative technologies with fast changes. In times of global warming we must also include the controversial issue of evaluating damages from greenhouse gases if choosing fossil alternatives. When evaluating and planning renewable technologies, engineering know-how is important, but insufficient. Since financing is a crucial issue for most renewable technologies with high front loaded cost and long amortisation periods, a thorough and trustworthy financial and economic analysis is necessary not only to avoid financial failure later on, but also to attract stakeholders like private investors, banks and government institutions to support a still unknown technology.

**Keywords:** renewable energy; tidal stream; photovoltaics; wind energy; hybrid system; financing; discount rate; social discount rate; greenhouse gas; CO2-price; ocean energy; PV; learning effects

**JEL Classification:** M2

## 1. Introduction

With increasing focus on global warming the attention on the use of renewable energy (RE) for electricity generation is rising. Some countries have already achieved levels of around 40% of electricity supply from renewable sources, mainly from wind and photovoltaics (PV), which have left their infancy stage. A hitherto much neglected technology is tidal stream energy (TSE), capturing energy from the current of the tides through underwater turbines. TSE has the potential to generate electricity for up to 20 h daily, and the availability of energy is precisely predictable even for years in advance, since the movement of the moon (and the sun) moving the tides is well known. This predictability makes it superior to intermittent technologies like PV and wind.

Tidal energy is a part of what we call "ocean energy", consisting of four types of electricity generation technologies: whereas tidal energy uses the flow of the daily tides, wave energy is using the

energy of waves, ocean thermal energy conversion (OTEC) the different temperatures on surface and sea ground, and salinity gradient takes advantage of the salt content of the water. Tidal energy and wave energy are the most used technologies, but even within the tidal system there are three types: traditional "tidal barrage" structures with huge dams near the shore, known from popular sites like the "La Rance Tidal Power Station" in France "Annapolis Tidal Generation Facility" in Canada, or "Sihwa Lake" in Korea, which is the largest station worldwide. All stations besides Sihwa Lake were built already around half a century ago, but considering their enormous size their capacity is rather small. The second type, TSE, is an advanced technology in much smaller scope, based on a floating device or sometimes ground mounted, having the dimension as well as the shape of an airplane or sometimes just a raft. Therefore the TSE-type can be applied in a much smaller setup than the huge barrage type, even in remote areas which are not connected to the main electricity grid. Turbines are small and in some designs they can be lifted out of the water for repair and maintenance, reducing operation cost.

A third type of tidal energy are tidal lagoons, which are similar to tidal barrages, although the distinction between a barrage and a tidal lagoon in technical terms relates primarily to whether an estuary is blocked (using a barrage) or whether the resource is captured using a barrier or seawall attached to two parts of the shore (Hendry 2016, p. 10).

The kilowatt-hour cost of TSE-electricity is still on the high side, but there are still chances for commercial application: on small islands or remote villages, frequently expensive and air-polluting diesel generators are the main source of electricity, so TSE could be a cheaper and environmentally better alternative. In other cases, where main demand comes from residential areas, the real life capacity-utilisation of other RE-technologies is not always optimal: capacity utilisation of photovoltaics is often as low as 15% so that the theoretical cost-levels can never be achieved (Rink 2017). TSE needs less backup capacity and has a much longer daily availability, so often the theoretically higher cost turns out to be lower in real life when considering total system operation and system integration costs. But although TSE is well developed, there is still a lack of long-term experience about reliability and maintenance requirements, making it difficult to find private stakeholders.

Therefore a proper economic as well as financial cost calculation, showing the cost curves and cash flows of different combinations of technologies for different locations with divergent characteristics is crucial for the success of such an innovative technology.

Therefore, three research questions shall be investigated:

1. Which combination of RE-technologies is the best to save cost and to contribute towards the decarbonisation of a remote place?
2. How can financial and economic methods help to calculate the true cost and to help reducing uncertainty for potential investors?
3. Can learning effects be expected to bring the cost down?

*1.1. Literature Review*

1.1.1. Technology and Quantitative Development

So far TSE is still a nascent application: Ocean Energy Europe, a non-profit organisation representing many professionals engaged in ocean energy research and industry, report in their 2020 report that there is an installed capacity of roughly 32 Megawatts (MW) worldwide, mainly in Europe (Ocean Energy Europe 2020). This is the equivalent of just a minor power station for a small city of around 10,000 households and just to put it into dimension: the worldwide installed capacity of wind plus PV is around 40,000 times higher (International Energy Agency (IEA) 2019). There are expectations that tidal stream energy will take off over the next few years. A market study (European Commission 2018) on existing and planned ocean-energy projects, asking 21 technology and project

developers[1], reported a total of 897 MW in 25 TSE-projects over the next few years, which would mean that existing capacity would rise 30-fold. Less capacity is planned for other ocean energy, with 111 MW in 16 wave-projects and 16 MW in one OTEC project. Being a predominantly European technology so far, only 67 MW of TSE-projects are planned outside Europe (p. 11). Three different scenarios based on a different percentage of projects delayed or cancelled are presented, also taking into account an earlier study from Magagna et al. (2016). For 2030, the optimistic scenario comes to a cumulative installed capacity of 2000 MW, the medium and pessimistic scenarios to 1600 MW and 1300 MW. Even the pessimistic forecast looks rather optimistic taking into account that the report (which is based on 2017 data) forecasted 200 MW (medium) and 80 MW (pessimistic) for 2019, in comparison to the actual 32 MW at the beginning of 2020. However, there is some hope that a progressive development sets in once more and more TSE-projects rise from the trial stage to commercial operation.

Lewis et al. (2019) also checked the variability of tidal energy streams on a TSE-turbine on the Orkney Islands and found that persistence and quality of the electricity provided is well in between usual tolerance levels and therefore TSE, other than PV and wind, can indeed be considered as highly reliable sources.

Kappatos et al. (2016, p. 286) state that the tidal power industry has made significant progress towards commercialization over the past decade and "significant investments from sector leaders, strong technical progress and positive media coverage have established the credibility of this specific renewable energy source". However, they still see that progress is being retarded by operation and maintenance problems and this will not improve as long as data obtained during industrial prototyping are kept confidential.

The next important step for tidal energy is to achieve learning effects as well as cost-reduction through the deployment of more demonstration projects (Magagna et al. 2016, p. 6). Ecorys and Fraunhofer (2017), in their study for the European Commission, conducted 57 interviews with stakeholders within the industry and they see TSE already out of the demonstration stage: "many tidal devices are already moving down the learning curve. The technology has converged in the basic design, so no major barriers are lying here anymore" (p. 27). Nevertheless there is a warning from too much enthusiasm: "The interview results show a clear consensus that sector-wide objectives have long been overambitious, resulting in a race towards commercial readiness, which incentivised developers to scale up too quickly. While investors were attracted, they pulled out again once they realised that the time to market turned out to be significantly longer than expected" (p. 28) The main problems are seen in installation as well as in finance problems: installations of equipment in the sea are challenging, but in addition the limited time window available to sink turbines and installations in areas with strong tidal currents (as little as 30 min) can be a major cost and risk factor as well as an important factor behind delays (p. 29).

### 1.1.2. Cost and Finance

A fact that should not be neglected is that the cost of electricity generated is still on the high end. In a quite common setup of around 2–6 MW installed capacity, the levelised cost of energy (LCOE)[2] comes to between 35 and 48 US-cents per KWh (Rink 2017), which is far beyond affordability for small households and also non-competitive for enterprises. We can expect that cost will increasingly come down as it did in the case of photovoltaics and wind, but only when TSE reaches a commercial operation stage of a significant number of plants learning can take place and cost can come down (International Renewable Energy Agency (IRENA) 2020). Coles et al. (2020), in a very recent simulation of a plant in Naru street in Japan with four phases, starting with 1 MW up to a total capacity of 45 MW came to the conclusion, that in the final phase of the array levelised cost of energy of £ 85/MWh could be

---

[1]    97 developers were contacted, but it can be assumed that the majority of non-respondents did not plan any ocean projects, since only about 30 were identified who are currently pursuing TSE-projects. The report concedes, however, that the coverage outside Europe has not been very high. The reason for this could be that either there are not many projects outside Europa, but also that developers are more secretive about their plans.

[2]    LCOE show the discounted cost of production divided by the discounted amount of electricity in MWh or KWh (see ch. 3).

achieved, which would mean 10.3 US-cent/KWh. The assumptions are not unrealistic, with a capacity factor[3] of around 30%, a technical availability of 95% and 10% losses, but it remains a simulation without a reality check. Therefore, when looking for a sustainable business model now, one has to rely either on subsidised feed in tariffs or, as Ecorys and Fraunhofer suggest, check for places like for example the Canary islands, where prices of electricity are high. This can also refer to South East Asian islands, where fuel prices for diesel generators are also high and therefore TSE technology as well as other RE-technologies can be already commercially competitive (Ahmed and Logarta 2017; Al-Hammad et al. 2015). Installing TSE in such places could, therefore, be a welcome opportunity to move down the learning curve and reduce cost.

The main obstacle to further unconstrained growth of TSE is clearly seen in finance. Ecorys and Fraunhofer (p. 59) define five levels of technical readiness: (1) Research and development, (2) prototype, (3) demonstration, (4) pre-commercial and (5) industrial roll out and they concede, that TSE is now just on the brink of the industrial roll out phase, after multiple demonstration projects, some pre-commercial projects and some projects which have already achieved private funding, and thus more advanced than wave energy. However, they see investors still being hesitant not only because of a lack of technological maturity, but because of the heavily front-loaded financing requirements, and, as it is standard for RE-projects, the long payback periods of at least 10 years. An overarching finance barrier lies, however, in the high risk levels, which under the Solvency II and Basel III rules are not classified as investment grade and, therefore, unavailable to institutional investors such as pension and insurance funds. Only once the risk profile for TSE decreases, this barrier is likely to be overcome (Ecorys and Fraunhofer 2017, p. 54).

The market study of the European Commission (2018) identifies the same obstacles: "long time horizons, complicated project planning, and coordination of multiple public- and private-sector partners make it difficult to structure deals". Due to its pre-commercial nature and/or unproven technologies, the ocean energy sector is usually too risky for market-based finance, and hence considered not "bankable". In addition, ocean energy projects may be considered too capital-intensive for venture capital investment and too risky for private equity financing. Of course, there might be investors willing to participate if there is a high return. However, looking at the high cost of TSE projects, their expected return does not match the prospects of venture capital and private equity (p. 46). Ecorys as well as the European Commission conclude, that just as in the case for more mature RE technologies, government support (e.g., by feed-in tariffs) and support by public finance is still necessary to get TSE commercially off the ground and to enable learning effects. Even on the islands with the high cost for diesel generated electricity, where a rather poor population is living, electricity use is mostly supported by subsidising the tariffs.

### 1.1.3. Reduction of Carbon Emissions

Any review on renewable energy would be incomplete if it did not consider the potential of avoiding greenhouse gases (GHG). It was Nordhaus, who was one of the pioneers suggesting prices for emissions: "Recall that carbon emissions are economic externalities—activities in which people consume things but do not pay the full social costs, and the single most important market mechanism that is missing today is a high price on $CO_2$ emissions, or what is called *carbon prices*." (Nordhaus 2013, p. 222). There are more greenhouse gases[4] than just $CO_2$, but in order to create a single indicator, the $CO_2$ equivalent ($CO_2$-eq) is used, which includes the carbon emission equivalent of all gases that

---

[3]　Capacity factor is a key ratio showing the actual use of the theoretical capacity, which will normally not be reached. If a power station has a capacity of 1 Megawatt (MW), its theoretical maximum output is 8760 MWh, which is 1 MW times the 8760 h of the year (365 * 24). If the actual generation in a particular year is 4380 MWh, the capacity factor is 50%.

[4]　Greenhouse gases are those gaseous constituents of the atmosphere . . . that absorb and emit radiation at specific wavelengths within the spectrum of terrestrial radiation emitted by the Earth's surface, the atmosphere itself and by clouds. This property causes the greenhouse effect: water vapour ($H_2O$), carbon dioxide ($CO_2$), nitrous oxide ($N_2O$), methane ($CH_4$) and ozone ($O_3$) are the primary GHGs in the Earth's atmosphere. Moreover, there are a number of entirely human-made GHGs in the

would cause the same integrated radiative forcing or temperature change over a given time horizon. (Intergovernmental Panel on Climate Change (IPCC) 2018, p. 546). The most difficult and the most controversial issue, however, is the price to be attached to a unit of $CO_2$. Nordhaus has integrated 13 different models targeting a maximum temperature change of 2.5° in half a century (Figure 1), which show a variation between 10–50 US-$ in 2020 up to between 30 and 200 US-$/ton-$CO_2$ in 2040.

The World Bank, in a study of the "High-Level Commission" (World Bank and IDA 2017, p. 3) concludes that the explicit carbon-price level consistent with achieving the Paris temperature target is at least US$40–80 per ton of $CO_2$ by 2020 and US$50–100/ton-$CO_2$ by 2030, provided a supportive policy environment is in place. The IPCC Report in 2014 (Intergovernmental Panel on Climate Change (IPCC) 2014, p. 79) looks at a couple of studies, in which the price per ton of $CO_2$ varies between a few dollars and several hundreds of dollars.

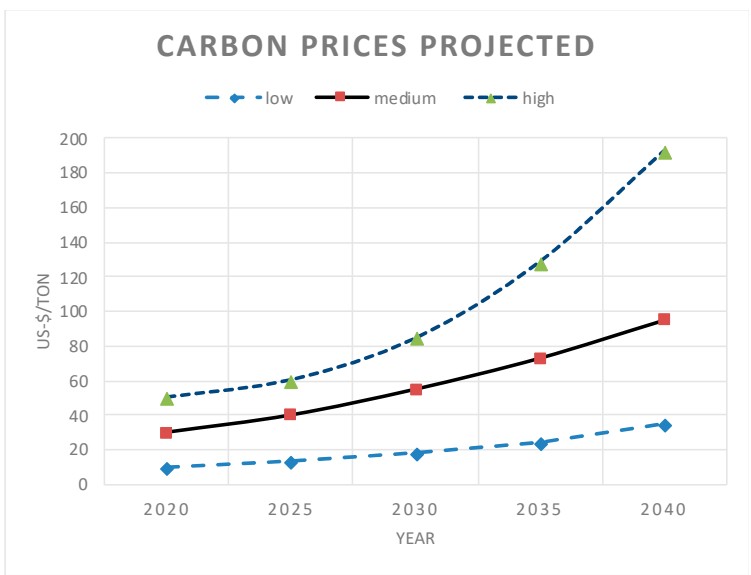

**Figure 1.** Different paths for $CO_2$ prices needed to limit temperature rise to 2.5 degrees Celsius.[5]

These values should reflect the "social cost of carbon", which are more or less arbitrarily fixed according to damages estimated and therefore often subject to the opinion of the evaluator. Alternatives are emission trading systems, which leave the termination of prices to the market. In the European Union (and some other countries) an emission trading system is in force, where the price fluctuated between 28 (in 2008) down to below 3 € in 2013 and around 20 €/ton in 2020 (Borghesi 2010, p. 12; Genovese and Tvinnereim 2019). Of course, these values also do not necessarily reflect the actual damages caused or avoided by a ton of $CO_2$, because demand and supply for such certificates depend on the amount of certificates in circulation and also on the particular business situation in $CO_2$-producing industries. So even with a working emission trade system prices fluctuate and a calculation of emission cost within a time frame of up to 20 years will include much guesswork. The consensus working rule in IPCC and United Nations Framework on Climate Change will also make it unlikely that the world will arrive at uniform carbon emission pricing in the near future (Weitzmann 2015).

## 2. Evaluation Methods for Renewable Energy (RE)

In RE as well as in TSE the usual financial and economic assessment methods are used, therefore it may not appear necessary to waste much effort on explaining them. In many appraisals from banks

---

atmosphere, such as the halocarbons and other chlorine- and bromine-containing substances (Intergovernmental Panel on Climate Change (IPCC) 2018, p. 551).

[5]   Adapted from (Nordhaus 2013, p. 228).

and financing institutions, however, methods like net present value (NPV), internal rate of return (IRR) or weighted average cost of capital (WACC) are used routinely without much consideration for their actual meaning, therefore it appears appropriate to recall their original sense and to consider whether their application is appropriate in a particular case. Since RE projects carry an extremely front-loaded (investment-) cost structure, there is an urgent need for finance over a prolonged time period, making reliable and exact financial assessment mandatory. In addition there is not much manoeuvring room for later changes and, therefore, a thorough evaluation at the beginning will also lay the foundation for an appropriate dimensioning of RE-projects, saving financial trouble later on.

*2.1. The Cost of Capital: Financial and Social Discount Rates*

Net present value (NPV), first discussed by Fisher (Fisher [1930] 1954), is the most fundamental valuation method in financial evaluation. The idea behind is that an amount of money received later has less value than an amount received now. In times of negative interest rates in some regions of the world we should probably define it more neutral: that money received later has a *different* value than received today (or earlier).

$$\text{NPV} = \sum_{t=0}^{n} \frac{Receipt\ stream_{(t)} - Outlay\ stream_{(t)}}{(1 + discount\ rate)^t}$$

For each period (*t*), mostly a year, the initial investment and the net benefits (receipt stream–outlay stream) are discounted to a present value. A positive NPV shows that the project is worth to be realised in principle, but it should also be compared to possible alternatives.

A decisive parameter is, of course, the discount rate, which will normally be defined as the *opportunity cost of capital*: if we invest money into a project, will this project yield the best return we can get or will we miss better opportunities? Higher discount rates will reduce later cash flows and vice versa. Renewable energy projects, for example, normally have a heavily front-loaded expenditure structure, which will make them look unfavourable against fossil energy projects, where fuel cost are coming later.

There is a discount rate for the *financial view* and a discount rate for the *socioeconomic view*. The financial rate will normally represent the market rate and include a risk component. Often the weighted average cost of capital (WACC) will be used, which will be discussed below.

There has been a major debate going on for many years about how the social discount rate (SDR) should be determined. A recent overview of the dispute provides Campos et al. and the most fundamental arguments and further sources can be found in Harberger and Jenkins (2002). In brief, SDR should represent the opportunity cost of (social) capital, which reflects the foregone benefits of the displaced resources from alternative uses that the society decides to invest today (Campos et al. 2015, p. 9). It will depend on the type of project how much SDR will differ from the financial discount rate. The more intangible benefits (like time savings, pollution reduction, health improvements etc.) will occur, the more difficult and sometimes subjective it will be to find a satisfying solution. Power projects, especially in remote areas, frequently trigger the question how to evaluate the benefits brought to the population to have electricity for commercial activity, reading and learning at night and using Wi-Fi and computers. In most cases such benefits will be just explained verbally without attaching quantitative values. Shadow prices, however, should be used whenever tariffs or fuel prices are distorted by taxes or subsidies.

International institutions like the World Bank use relatively high social discount rates of around 10% for developing countries, the European Union between 3 and 5% within Europe. Developing countries like India or the Philippines are using also higher rates between 10% and 15% for local projects (Campos et al. 2015, p. 31). In these countries it must also be considered that we often have higher inflation rates and weak exchange rates, thus when starting an assessment it should be clear what currency is used and whether the calculation is in real or nominal values. This is particularly

important if income mainly comes from local sources (as is normally the case with power projects), but debt and interest goes to foreign banks and shareholders.

There is also a debate going on among environmentalists on discount rates, since a high discount rate will considerably reduce damage caused by global warming in the far future (Stern and Stern 2006). Some also suggest "hyperbolic discounting", which reduces the discount rates with time, thus assigning less reduced values to figures in the far future (Nordhaus 1999; Goodin 1982). For decisions with a very long time horizon, especially when assessments on global warming are concerned, such an approach should definitely be considered.

If we just want to compare different projects which yield the same income, a simpler formula for present value of cost (PVC) instead of NPV can be used, just taking into account the discounted cost of the project.

The internal rate of return (IRR) is the interest rate that equates the present worth of a cash flow stream to zero (Turvey 1963). If we have problems finding the right discount rate, this will be a helpful solution. The IRR calculation assumes that benefits from the project under review are re-invested at the internally generated rate of return, yielding further benefits in the next period. This might not be possible, but it will significantly distort the results only if IRRs are very high. Under such circumstances the World Bank suggests a modified IRR (MIRR), which "corrects for this by assuming benefits are re-invested at the opportunity cost of capital" (World Bank 2005, p.4).

## 2.2. Weighted Average Cost of Capital (WACC)

Financing a project requires either equity or debt or, mostly, both. WACC answers the question. which are the average cost for a firm or for a project under given proportions and given interest rates for any combination of both sources of capital.

$r_c$ = WACC = (return on equity $\times$ share of equity) + (interest rate of debt $\times$ share of debt)[6]

The result can also be used to determine a meaningful (financial) discount rate. Since normally debt will be cheaper than equity, in theory WACC can be minimised by increasing leverage in favour of debt. However, banks will rarely agree to very high shares of debt exceeding 80% (sometimes lower) and, in addition, loans with high leverage will carry higher interest rates.

For determining the required return on equity the Capital Asset Pricing Method (CAPM) formula, which was first presented by Sharpe (1964) and Lintner (1965), is common practice:

$$\text{return on equity } (r_e) = r_f + \text{ß}(r_m - r_f).$$

$r_f$ = risk free interest rate
$r_m$ = market rate (required or estimated return)
ß = coefficient of the systemic risk of the firm or the project

The decisive parameter in this formula is Beta, which defines the relative risk of the firm or the project in relation to the market risk, by using the correlation coefficient of the firm's shares with the market and the respective standard deviations (Sharpe 1964; Lintner 1965). A Beta greater 1 indicates that the risk of the individual firm or the venture project is considered higher than the market risk and vice versa. For listed companies, these values can be obtained by looking at the fluctuation of the share prices versus the market and these data are normally easily accessible from financial data providers. For non-listed companies we can use similar listed companies for approximations, but for innovative projects like TSE there are no reliable standard deviations available.

---

[6] Frequently an "after tax" formula is used, which considers that interest for debt will be part of the cost and therefore reduce taxes. Here the debt share is corrected by the tax rate $(1 - t)$. It depends, however, on the concrete situation of the project as well as on the tax regulations if WACC after tax is more suitable than the so called "plain vanilla-WACC".

For TSE we can refer to data from renewable energy companies or better even, solar and wind projects. Rezec and Scholtens (2017) have examined the market performance of 14 listed RE-enterprises between 2000 and 2013 and their regression analysis shows that only one company has a Beta below one. Twelve others are significantly above one with values between 1.3 and 1.8. One case shows a non-significant result (p. 372).

The result is not really surprising, but it has to be considered that these are already established companies and no start-ups investing into a nascent technology.

A very recent study from Steffen (2020) analyses the performance of mainly wind and solar projects documented in accessible publications, investor surveys and company reports in 33 countries for wind projects and in 23 countries for solar projects. Steffen does not investigate Beta, but in a straightforward way the weighted average cost of capital and the results differ by year, country and technology. In all countries and with all technologies there is a downward trend with time, which can be explained by their increasing maturity. The highest WACCs are shown in Greece, India, Guatemala and Thailand (12–15%), the lowest in Germany, Denmark, but also the United Arab Emirates, Saudi Arabia and Peru (2.5–5%). There is also a surprising split within third-world countries: high values above 10% in Cambodia, India, Malaysia, Thailand and Vietnam, values below 5% in Jamaica, Peru, Saudi Arabia and Zambia. However, it has not been further investigated whether the causes for the higher values are caused by country risk or maybe other internal facts like tariff structures or (missing) government guaranties or support. There are also rather high values above 10% (mainly for wind) in European countries like Croatia, Bulgaria, Romania, Slovenia and Hungary. Technology-wise the lowest WACCs are shown for solar. Slightly higher ones for on-shore wind and the highest ones for off-shore wind.

Although we can use these data to estimate a suitable WACC and thus a suitable financial discount rate, we should be aware that they remain estimates. Even the CAPM-calculations for listed companies are not necessarily a reliable predictor for the future of these companies. Which WACC we use also depends on the purpose: if we are looking for financing sources. the rates should be close to the technology (e.g., off shore wind for TSE) as a starting point, but if we are calculating the NPV for a TSE project and compare it with a base case of fossil fuels or other RE-technologies. We cannot use two different values and thus we must compromise and use a mixed discount rate.

*2.3. Levelised Cost of Electricity (LCOE)*

Levelised cost of electricity (LCOE) or levelised avoided cost of electricity (LACE) are standard parameters for measuring the long term cost of electricity (EIA 2020; Roth and Ambs 2004):

$$LCOE = \frac{\sum_{t=o}^{n} \frac{I_t + M_t + F_t}{(1+r)^t}}{\sum_{t=0}^{n} \frac{E_t}{(1+r)^t}}$$

$I_t$ = Investment cost
$M_t$ = Operation and maintenance cost
$F_t$ = Fuel Cost
$E_t$ = Electricity generated (minus losses) in KWh or MWh
(all values in period "$t$" = time)
$r$ = discount rate

LCOE indicates the average cost of building and operating a generating plant during an assumed financial life per unit of electricity. Key inputs include capital and fuel costs, fixed and variable operations and maintenance (O&M) costs, financing costs and an assumed utilisation rate for each plant type. The relevance of each of these factors varies across technologies: for RE-technologies the capital costs dominate. LCOE will also, like NPV, change in relation to the discount rate used: for technologies with high initial (capital) cost the discount rate will not make so much difference as for technologies with high fuel cost and therefore a high discount rate will favour fossil technologies over RE.

　　　LACE (levelised avoided cost of energy) is used when comparing a "base case" with a new investment to determine how much per unit of electricity will be saved by the new technology which replaces the old one. LACE is not just a figure like LCOE, but a methodology: it requires a wider consideration of effects by also looking at avoided cost and in particular the difference in cost of LCOE between two technologies from a system view.

　　　For an economic analysis, using the social discount rate, externalities like the monetary value of $CO_2$ emissions and potentially other damages may also be included in a comparative analysis (Roth and Ambs 2004). It should also be noted that LCOE does not just include the cost of a single plant, but also the system cost necessary to integrate the envisaged plant into the entire electricity grid. If integration cost and cost for providing backup facilities as well as cost for overproduction at times of low demand are not included, the result might be too much in favour for RE technologies (Ueckerdt et al. 2013, p. 74).

## 3. The Project

### 3.1. Suitable Locations

　　　Between 2017 and 2019 a pre-feasibility study was carried out investigating technical, financial and economic preconditions for TSE- projects on the Philippine islands. There were plans to continue this study as a full feasibility study by the second half of 2020. In 2019 site visits were carried out and the idea also found official support from the Philippine Department of Energy (DoE). Information was also collected mainly from two manufacturers and consultant engineers active in the field of TSE. One of the manufacturers, Schottel Hydro, together with a Singapore-based consultant provided valuable data on the suitability of several locations in the Philippines and other South-East Asian countries.

　　　The Philippines together with Indonesia and Myanmar have many promising project sites as there are many narrow straits between the islands. The Philippine Department of Energy (DoE) estimates the theoretical potential of ocean energy in the Philippines at 170 GW (Figure 2). Twenty indicative sites for TSE have been defined in the country.

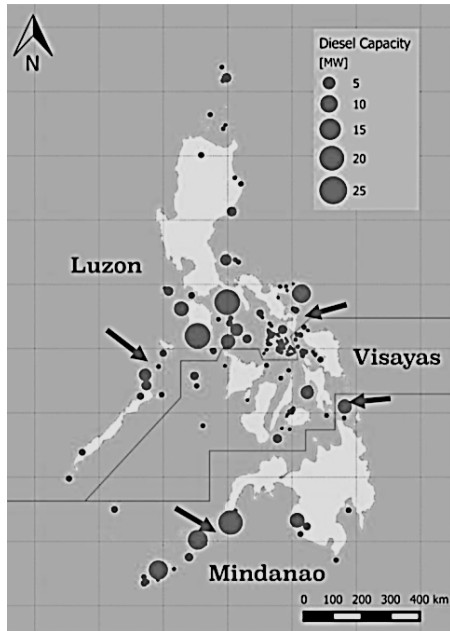

**Figure 2.** Potential locations replacing diesel generators in the Philippines.[7]

---

[7]　Source: (Lochinvar and Sim 2018).

TSE is very suitable for providing electricity to "off-grid" areas such as small islands. Such areas are presently mostly served by mini-grids, powered by diesel and bunker oil-powered generators. Generation costs can reach 0.88 USD/kWh (average 0.45 USD/kWh). These islands suffer frequently from blackouts and unplanned power outages. Only 22 of 233 areas have 24/7 h electricity, with over 70% having less than eight hours per day of electricity. An installation and continuous operation of several TSE-projects in such areas would not only improve the electricity supply for the population on these islands, but it would also facilitate a learning process which causes cost to come down as it was the case for PV and wind. Finally, TSE could also be a valuable and clean source of electricity generation on other islands as well as on the mainland.

## 3.2. Electricity Tariffs

Financial sustainability of any project requires cost-covering revenues, which is harder to achieve when the production cost are above international standard. In order to assure affordability for the rural end-users. The tariff is subsidized down to a level of around 10–12 US Cents per KWh by a fund called UCME (Universal Charge for Missionary Electrification). This fund is cross-financed from contributions of all users connected to the main grid (Ahmed and Logarta 2017). If we manage to install a TSE system generation electricity for less than 45 Cents/kWh on the average, the subsidies to be paid could be reduced considerably, although the tariffs will not cover the cost in full.

In addition substantial pollution could be avoided and electricity would be provided in a reliable way and 24 h per day, if storage possibilities are included.

## 3.3. Hybrid Tidal Stream Energy (TSE) Systems

The first calculation was made on Dinagat, part of Dinagat island northeast of Mindanao, with several model alternatives, using the "Hybrid Optimisation of Multiple Electric Renewables" (HOMER-Pro) software, which can analyse different combinations of generating systems. The energy supply costs are minimised while ensuring that demand is matched on an hourly basis. If hourly grid demand data are not available, HOMER-Pro provides the option to generate synthetic demand profiles depending on the power system size and its economic activities (Rink). Further calculations were made on Rasa island and a few other minor spots. Since island-specific peculiarities should not distort the picture, the calculation was later transferred to a hypothetical model island containing average data from several locations. In an island network without the possibility to use electricity from a nationwide grid, TSE still needs either a storage system or a second-generation system if a reliable supply for 24 h in seven days is required. The theoretical alternative of a 100% PV-system (supported by storage) turned out to be too expensive because of its very low capacity factor in an area with a predominant demand of electricity at night. On the other side a 100% TSE system (plus storage) also proved to be more costly than a combination of TSE and PV, supported by battery storage, which was found to be the optimal way from the cost side as well as from the operation side for most locations.

Our base case, averaged from existing locations, is a 5 MW diesel plant with older diesel generators which would have to be refurbished immediately to be fit for a 24 h operation. In our base case it is assumed that the machinery will have to be replaced with a specific investment cost of 462 US-$/kW after five years and also 10 years afterwards. A moderate price increase of 10% (which is 1.95% p.a.) was assumed for year 15, but a residual value was also assumed for year 20, the expected end of economic life for the RE alternative.

In times of high demand, the charging of batteries will require excess capacity above peak load. Therefore the alternative chosen is a hybrid combination of 9 MW (4.5 MW TSE and 4.5 MW PV) capacity, based on TSE, PV solar and battery storage. With an envisaged peak load of 4 MW a storage capacity of 20 MWh would be able to provide electricity for five hours in off-times, average load up to eight hours. which is actually more than sufficient and accommodates some growth, but the exact sizes of storage as well as of generating capacity would be determined during detailed project planning, which also must include a thorough survey of future demand and potential new customers. The

capacities assumed in our model case are generally on the high side in order to avoid accusations of bias in favour of RE technology. For details see the appendices.

### 3.4. Financial Evaluation

In order not to create too much confusion the variety of alternatives was strictly limited to the two aforementioned alternatives

1. The *base case*, being the continuation of existing diesel stations (which is the case on more or less every island) with some improvement to reach a 24/7 service.
2. The *hybrid combination* of TSE, PV and battery storage.

Some variations will be briefly outlined in the discussion part. The core data can be found in Table 1. Because after the Corona pandemic crude oil prices fell, a lower assumption was made for the diesel price. However, we do not predict that this lower value will pertain for 20 years, so we also used the previous price of 60 pesos/L (1.20 USD), which was the price before on most of these islands. A diesel-price of 52.15 pesos (1.043 USD) marks the break-even point between the base case (diesel) and the hybrid solution, any price above will favour the hybrid alternative. The (financial) discount factor was set at 10%, in line with the expected financing cost.

The hybrid alternative requires a much higher investment cost which makes financing a challenge and which will require a loan, whereas the anticipated re-investments of the base case can be made out of the generated cash-flow, provided the subsidized tariffs pertain. On the other side, the operation cost (OPEX) of the hybrid solution are much lower than the OPEX of the base case.

The LCOE of the hybrid case will be 27.4 US cents per KWh, in the case of the diesel alternative they range between 27.4 cents (for the break-even diesel price) and 36.1 cents, more likely on the high side unless oil prices remain low. In both cases the hybrid-alternative is equal or better unless there will be a further decline of crude prices.

**Table 1.** Core values of base case and hybrid alternative.

|  | **TSE, PV and Batteries** | **Base Case (Diesel)** |
|---|---|---|
| Generation mix | 4.5 MW TSE Generator; 4.5 MW Solar PV | 5 MW Diesel generator |
| Electricity Demand | 22 GWh p.a.; Peak Load: 4 MW. System capacity factor: 63%. average load 2.52 MW | |
| Storage: | 20 MWh Batteries Storage | - |
| Investment costs | 36,765,000 USD (Year 0) 2,400,000 USD (Battery replacement Year 10) 39,165,000 USD | 825,000 USD (refurbishment) 2,310,000 USD (after 5 years 2,541,000 USD (after 15 years) 5,676,000 USD [a] |
| Opex Cost p.a. (average) | 1,666,900 USD p.a. | 7,665,000 USD p.a. |
| Assumed PPA tariff $ | 0.309 USD/KWh [b] | 0.451 USD/KWh |
| LCOE (r = 10%) | 0.274 USD KWh | 0.274 USD/KWh (Diesel 1.04 $/l) 0.361 USD/KWh (Diesel 1.20 $/l) |
| Additional GHG-emissions | - | 16,880 t resp. 13,129 $CO_2$-eq/p.a, after 5 years |

[a] Assumed residual value in year 10: 1,155,000 USD. [b] The tariff was arbitrarily fixed at a rate that allows a return on equity of 15%, deemed necessary to find investors. MW(h)= Megawatt (hours), GWh= Gigawatt-hours, GHG= greenhouse gases.

In both cases the actual tariffs of around 10 US cents are much lower than the LCOE; at present more than 45 cents are paid to the power producer for the base case, which would provide a substantial

profit under the assumed circumstances. Because of this, the government has been discussing for quite some time whether to reduce these so called UCME-subsidies. Therefore, for the hybrid case we assumed a rate that is just sufficient to reach an equity rate of return of at least 15% with an equity share of 30%. Here we arrive at a much lower "UCME-tariff" of 30.9 cent/KWh, which would save around USD 3.1 million per year just for one location. Profit tax is not included in the above calculations, since it is very likely that tax exemptions are granted for these missionary islands. If a tax of 10% (regular missionary islands rate, KPMG 2018) were applied (Appendix A), the ROI would still reach 14.2%, but with a tax of 30% still 12.3% would be achieved. In this unlikely case this could be compensated by an increase of the PPA tariff by 10%.

*3.5. Financing*

The general problem of most RE technologies are the high initial investment cost, and in addition TSE is a still unknown technology. In this special case, a large Philippine public finance institution declared its readiness to support the project in the framework of a joint financing of several financing partners. The details will have to be worked out once the detailed project planning is completed and the tendering process initiated. But for the time being an equity of 30% was assumed and the UCME tariff is set at a rate that enables a return on equity of 15%. Cash flow will be positive throughout the life of the project and the loan is envisaged to be repaid in 10 years (details in Appendix A). With a present UCME-tariff being much higher there would still be some manoeuvring room upwards in case the investment is found too risky for investors. On the other hand with the participation of a public institution and some potential green funds from the government a return of 15% does not seem to be undervalued.

*3.6. Socio-Economic Evaluation*

The following target groups will benefit:

- The residents of the island because of reduction of pollution and noise;
- Electricity consumers and small businesses from a more reliable 24 h supply of electricity;
- Philippine electricity consumers from reduced subsidy requirements;
- Local grid-operators (distribution utilities, cooperatives) through predictable grid management and secure supply;
- Project developers from new options to extend hybrid RE systems;
- Employees in construction and maintenance of hybrid RE systems as well as indirect job creation through more reliable electricity in business and tourism.

Such benefits are difficult to quantify, therefore we will not try to do this. But because of the ongoing discussion on global warming we should at least try to quantify the reduced greenhouse gas emissions. We estimate the annual reductions of $CO_2$-equivalent[8] to be 16,680 tons p.a. during the first five years and after the installation of new diesel generators in year 6 to be 13,129 tons p.a. For a pure economic calculation we must also deduct taxes, so we have to cancel diesel taxes (6 pesos per litre and −12% VAT) and also VAT for investments as well as import duties for the RE equipment. The result can be seen in Appendix B. Even without consideration of greenhouse gas effects the hybrid alternative (LCOE 21.51 cents/KWh) is still slightly better than the diesel alternative with (21.99 Cent/KWh).

If we consider the greenhouse gas effect, which we should, the result depends on the price we attach to the $CO_2$ emissions: since the hybrid solution fares better even at a price of zero, we do not have to start discussions about which price is appropriate. But as can be seen in Figure 3 the difference will become much larger the higher the $CO_2$ price will be.

---

8　Please be reminded that the $CO_2$-eq estimates the "warming equivalent" of all greenhouse gases.

At a price of 60 USD/t. which is the mean value of the medium scenario (between 2020 and 2049) in the Nordhaus-scenario (Figure 1), the (socioeconomic) LCOE of the diesel alternative reaches 26.6 cent/KWh (+24%). It should be noted that these figures refer to the socioeconomic analysis, which is net of taxes. It can also be applied to the financial analysis with the same effect, but in the real world a $CO_2$ price will only be effective if there is an emission scheme which requires the polluters to pay.

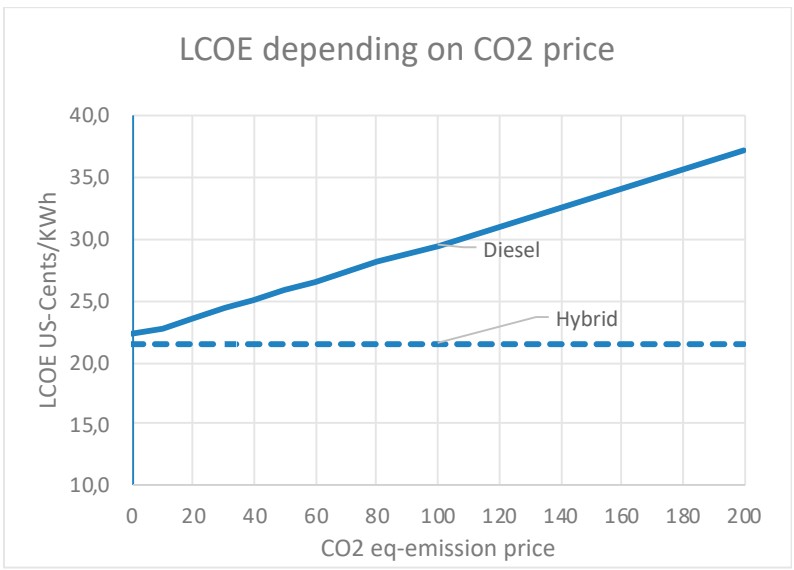

**Figure 3.** Levelised cost of energy depending on $CO_2$-eq price (socioeconomic calculation).

Considering savings of altogether 281,000 tons of $CO_2$-equivalent over the lifetime of just one installation on one island will probably be enough to convince even sceptics.

### 3.7. Environmental Impact Assessment (EIA)

When we talk about the environment we also have to see critical aspects: the sea bed conditions, coral reefs, the use of project sites for fisheries and transport, the exposure and access. For selected sites, detailed surveys are needed also regarding flow speed, flow direction, flow divergence, turbulence, depth, bathymetry, bed steepness, bed type etc. Once a tidal turbine is installed, new issues might be detected during its active life. However, there will be no dams nor barriers and the rotors are slow moving, which will cause less drastic changes on marine ecological factors. Experience so far shows, however, that the greatest change from baseline conditions occurs in the installation phase, but the extent of this change is reduced later on (European Marine Energy Center (EMEC) n.d.; Norris 2009).

## 4. Discussion

### 4.1. More Alternatives

Frequently, in locations selected for a potential TSE investment, diesel stations which are still in reasonable shape are running and thus it might make sense to let them work until their life expires. This, however, depends on their availability and the possibility to provide a 24 h service, which is mostly not the case. In order to reach a 24/7 service it can be a worthy solution to supplement them with TSE first and add PV and batteries later, when their time comes. If replacement takes place after five years, the investments for PV and batteries can be postponed, but due to logistics and installation cost a split of TSE into two phases would be uneconomical. With these assumptions (other values remaining unchanged) we would arrive at LCOE of 29.1 US cents/KWh, which is more expensive than our original TSE solution and even more expensive than the base case solution with 24-h diesel generators running. However, this refers to our model case: in real life settings, for example when the

existing generators are in good shape to last longer and without need for refurbishment to provide a 24 h service, this solution might still be preferable to an immediate change. Environmentalists might protest, but if one compares the negative side of more GHG emissions with the need to demolish and scrap physical installations that are still working, one might come to a different view. In any case it makes sense to include this possibility into the detailed project planning of each of the locations ready for such a project and of course the interests of the local providers and their co-operation must also be taken into account.

### 4.2. The Influence of Discount Rate

In our assessment we were using a constant financial as well as economic discount rate of 10%. In Section 3, the various opinions about the appropriate size of discount rates were discussed. In particular environmentalists criticise high rates because they find that such rates will undervalue long-term effects. It is clear that our front-loaded TSE investment will fare much better with lower discount rates compared to fossil alternatives with higher cost (and damage) later on. Since we are not in the position to determine an absolute correct rate, we shall just compare the results of different rates on our project (Table 2).

The hybrid alternative will benefit massively from a discount of zero, but also 5% the LCOE will still be far better than the base case. This refers to both the financial as well as the socioeconomic evaluation. If we increase the discount rate to 15%, however, the result will turn.

**Table 2.** Levelised cost depending on different discount rates (in US cents/KWh).

| Discount rate | 0% | 5% | 10% | 15% |
|---|---|---|---|---|
| Base case (Diesel) | 20.7–23.9 | 26.3–30.1 | 27.4–36.1 | 28.3–32.3 |
| Hybrid Case | 16.4 | 21.3 | 27.4 | 34.3 |
| Socioeconomic evaluation | | | | |
| Base case (Diesel) | 22.6–24.0 | 23.2–24.7 | 22.0–25.5 | 22.6–26.2 |
| Base case plus 25 $/t CO2-eq. | 22.1–25.9 | 23.2–26.6 | 23.9–27.4 | 24.5–28.0 |
| Hybrid Case | 13.3 | 17.0 | 21.5 | 26.7 |

However, as far as the financial discount rate is concerned, looking at the financial market and the respective risk a discount rate of 10% will be more adequate than 5% or even 0%. For fixing the socioeconomic discount rate we cannot rely on a market and most decisions in planning institutions and donors like the World Bank are based on estimates. Since in our case, however, the hybrid alternative fares better with all discount rates below 11% (including $CO_2$ emissions even below 13%) it appears safe to assume that the hybrid alternative shows economically a better result than the fossil solution

### 4.3. Other Parameters

There are two more parameters having decisive influence on the result: the diesel price and the $CO_2$ price when we compare our hybrid solution with a fossil base case. We have assumed a diesel price of 52.1 pesos (1.04 USD) per litre, which marks the break-even point in the financial calculation: In any place where the price is higher, and in fact on quite a few islands where the price goes up to 70 pesos, the hybrid solution will be better. In spite of the high front loaded cost there is one decisive characteristic of RE technologies: they are independent of price changes in the future, which can be a blessing, but also a curse if crude prices will fall.

The $CO_2$ price might be a strong instrument against global warming, but for the financial analysis, which is decisive for attracting investors, it is merely a fiction as long as there is no real life emission tax or certificate system in force. Only if fossil fuels will become more expensive will it influence the price and hence the decision of investors. Although China announced an emission trading system

in 2017 (Yang et al. 2016; Hubler et al. 2014) and in some provinces it is already in force, presently there is no such system really in sight for the East Asian region. For international as well as national institutions granting concessional loans or guarantees, however, avoidance of greenhouse gases might be an important argument and the participation of such institutions might in turn be an important precondition for private banks and investors to come into the boat.

Because a cost comparison of technologies was in the main focus, we concentrated on LCOE and not on NPV and IRR. NPV and even more IRR will be very important if the project wants to attract investors. The equity-IRR of the hybrid project is 15% (total capital 12.6%) and quite attractive, still based on the assumed reduced tariffs of 30.9 instead of 45.1 cents at present (Appendix A). Since this can only be reached on the basis on subsidised tariffs, it would be very important in the phase of detailed project planning to make sure that this subsidy will still be granted during the coming 20 years.

### 4.4. Learning

There is no proof that cost for TSE will come down once the technology is applied in greater scope, but experience with other RE technologies show that cost went down considerably in the last decade (and also before). Solar PV showed the strongest decrease with 82% cost reduction since 2010, in onshore wind cost fell by 38%. In offshore wind it fell by 29% (Figure 4). Even storage system prices have fallen, for example lithium batteries by 73% between 2010 and 2016 and they are projected to fall further (International Renewable Energy Agency (IRENA) 2017). All technologies are now in the cost range of fossil and nuclear technologies, the only problem is the still low capacity factor of solar PV, which rose only from 14% tom 18%, whereas both wind systems are in the region of 40%, however, with considerable local differences.

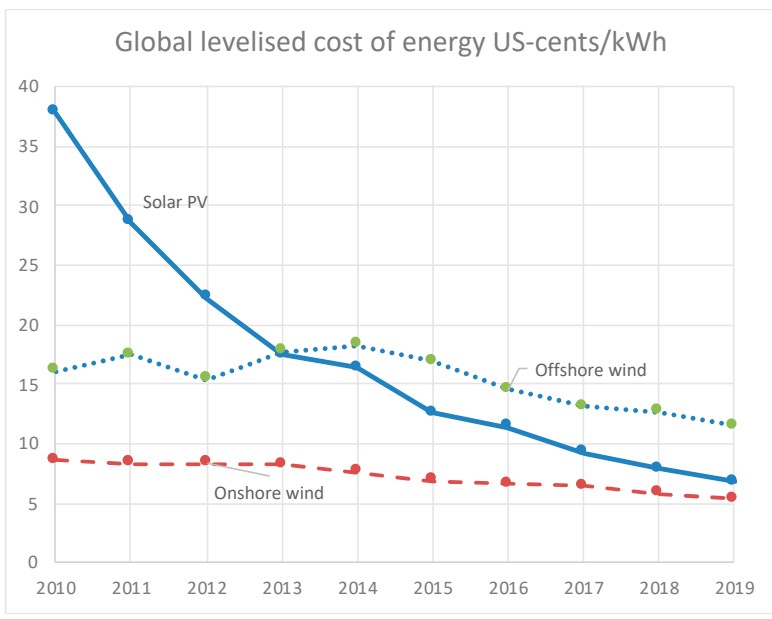

**Figure 4.** Global levelised cost of different renewable energy (RE) technologies 2010–2019[9].

In our case we can reach financial LCOE of 27 cents/KWh with a hybrid system of dominant TSE, which is where solar PV was in 2011. Since improvement of technologies by learning is to be also expected for TSE, we can hope for a price decrease to around 15 cents in five to six years. But it needs a great movement forward and there must be a significant increase in installed capacity. If we follow the

---

9    Adapted from: (International Renewable Energy Agency (IRENA) 2020).

European Commission's forecasts of at least 1300 MW in 2030, we should not worry about this. If this is achieved, TSE can also be operated not just on an island with high generation cost, but also in other suitable areas and they can be a welcome supplement to PV with its low-capacity factor.

## 5. Conclusions

Regarding our initial research questions we can conclude:

- A *hybrid combination of TSE with photovoltaics and supported by storage facilities* proved to be the most suitable solution for a pollution-free and economically superior alternative to the present diesel-operated generators in remote areas where present prices are still high. This result holds unless rather extreme conditions are assumed, like constantly low crude oil prices or a very high discount rate. The exact dimensioning of this hybrid solution must be refined during detailed project planning and adapted to the situation of particular locations.

- Usual *financial and economic appraisal methods* can be used, but it should be noted that even *mathematically exact calculations results can be deceptive and open for manipulation:* the discussion about the market risk (Beta) of RE companies and projects showed that there are no reliable empirical data available yet and even if there are, conditions are changing too fast to make past data really a trustworthy predictor of the future. *Discount rates will have a decisive influence on RE-investments* in comparison to fossil technologies and, therefore, results can easily be inverted by changing discount rates, even if they remain within a conventional range. Hence, when looking at an appraisal result of RE-projects we should always have a very thorough look at the parameters used. Since financing of RE-projects with usually long payback periods and front-loaded cost remains a crucial issue, utmost care must be taken to arrive at appraisal results which are trustworthy and at the same time appealing to potential investors.

- An open question is the *measurement of pollution*. As long as there is no emission trading system or a greenhouse gas (GHG) tax set by the government, the calculation of $CO_2$ or GHG prices remains an academic exercise which makes little impression on private investors. But for project developers this issue remains important, because governments and international financing institutions put an increasing emphasis on environmental aspects and require investors to present an analysis of ecological consequences if asking for public finance or support. Often banks do require such assessments too in order to avoid later claims for damages. In the long run, especially if we prefer market solutions, an emission trading scheme would be preferable to government interventions.

Solar as well as wind projects have shown considerable learning effects and subsequent cost reductions during the last decade. Therefore, there is no reason not to expect such learning for TSE and other ocean energy projects, provided there is sufficient opportunity for learning. As long as costs are still high, TSE requires (like other RE technologies before it) either governmental support or, as in our case, locations where high cost does not play a restrictive role, because existing alternatives are even more expensive. We have shown that many such locations exist and that hybrid TSE-solutions can contribute not only to a more reliable supply of electricity, but also to a massive reduction of pollution in these areas. It can be expected that after a couple of years of operation learning effects will reduce cost and make TSE a commercially viable alternative to other RE technologies, providing a longer and more reliable supply of energy than wind and sun.

**Funding:** This research received no funding.

**Conflicts of Interest:** The author declares no conflict of interest.

## Appendix A

**Project results: financial evaluation**

### 1. Base Case (Diesel) — Baseline

| | old | new**** | | | old | new |
|---|---|---|---|---|---|---|
| Thermal efficiency of Diesel | 35% | 45% | Assumed Fuel Cost for Diesel (USD/kWh) | | 0,297 | 0,232 |
| Capacity/Pek load MW | 5 | 4 | Assumed PPA-tariff ($/kWh)=end-user tariff+subsidy | | 0,451 | |
| Specific fuel consumption (l/kWh) | 0,285 | 0,222 | Life cycle period of investment (years) | 5 years remaining diesel, 10 years new diesel, | | |
| Price of Diesel (PHP/l) | 52,15 | | O+M Cost Diesel: | 5% of CAPEX | | |
| Price of Diesel ($/l) | 1,0430 | | | | | |

**Pay back scenario (baseline: Diesel only)**

| year | 0 | 1 | 2 | 3 | 4 | 5 | 6 | 7 | 8 | 9 | 10 | 11 | 12 | 13 | 14 | 15 | 16 | 17 | 18 | 19 | 20 |
|---|---|---|---|---|---|---|---|---|---|---|---|---|---|---|---|---|---|---|---|---|---|
| MWh sold*** | | 22 075 | 22 075 | 22 075 | 22 075 | 22 075 | 22 075 | 22 075 | 22 075 | 22 075 | 22 075 | 22 075 | 22 075 | 22 075 | 22 075 | 22 075 | 22 075 | 22 075 | 22 075 | 22 075 | 22 075 |
| Sales income (1000 USD*) | | 9 955,9 | 9 956 | 9 956 | 9 956 | 9 956 | 9 956 | 9 956 | 9 956 | 9 956 | 9 956 | 9 956 | 9 956 | 9 956 | 9 956 | 9 956 | 9 956 | 9 956 | 9 956 | 9 956 | 9 956 |
| - re-investment/res.value (1000 USD**) | -825,0 | | | | | 2 350 | | | | | | | | | | 2 585 | | | | | 1 155 |
| - diesel fuel cost (1000 USD) | | 6 562,0 | 6 562 | 6 562 | 6 562 | 6 562 | 5 111 | 5 111 | 5 111 | 5 111 | 5 111 | 5 111 | 5 111 | 5 111 | 5 111 | 5 111 | 5 111 | 5 111 | 5 111 | 5 111 | 5 111 |
| - Other O&M cost (1000 USD) | | 117,5 | 118 | 118 | 118 | 118 | 118 | 118 | 118 | 118 | 118 | 118 | 118 | 118 | 118 | 118 | 118 | 118 | 118 | 118 | 118 |
| Total Cost (1000 USD) | -825,0 | 6 679,5 | 6 679 | 6 679 | 6 679 | 9 029 | 5 229 | 5 229 | 5 229 | 5 229 | 5 229 | 5 229 | 5 229 | 5 229 | 2 644 | 5 229 | 5 229 | 5 229 | 5 229 | 4 074 |
| Cash Flow (1000 USD) | -825,0 | 3 276,5 | 3 276,5 | 3 276,5 | 3 276,5 | 926,5 | 4 727,0 | 4 727,0 | 4 727,0 | 4 727,0 | 4 727,0 | 4 727,0 | 4 727,0 | 4 727,0 | 7 312,0 | 4 727,0 | 4 727,0 | 4 727,0 | 4 727,0 | 5 882,0 |

* ) Sales income is irrelevant for LCOE

** ) Existing diesel stations continue for 5 more years. Refurbishment for 24 hour supply in year 1, specific inv. cost 470 USD/kW
replacement in year 5 and year 15, each 5 MW, residual value in year 20: € 1,050,000

*** )Average Load: 2,52 MW; Peak Load: 4 MW; System Load Factor: 63%

****) new diesel generators from year 6 onwards have a higher efficiency

| | | |
|---|---|---|
| discounted MWh | 187 939 | LCOE -274,1 USD/MWh |
| discounted cost | -51 509 | NPV (1000 USD) 33 251 |

ROE: not applicable, because investment amounts in the past are not known

| | Total | Total discounted | |
|---|---|---|---|
| MWH sold | 441 504 | 187 939 | |
| Sales income | 199 118 | 84 760 | 1000 USD |
| Total cost | -111 266 | -51 509 | 1000 USD |
| Cash flow | 87 852 | 33 251 | 1000 USD |

### 2. Supply mix: 4,5 MW TSR + 4,5 MW PV +20 MWh battery-storage

| | 1000 USD | Exch. rates EUR/USD: | 0,909 | | | specific 1000 USD | CAPEX 1000 USD | OPEX 1000 USD | Comments |
|---|---|---|---|---|---|---|---|---|---|
| Investment (1st year). =Capex 2 | 36 765 | | | | | | | | |
| Credit volume | 25 736 | Phil. Peso (PHP) | 50 | PHP/USD | TSE: | 6 600 per MW | 29 700 | | Capacity Factor: 40,0% |
| Equity | 11 030 | Annuity2 a = | 4188,33 | 1000 USD/p.a. | OPEX: 5% of CAPEX | | | 1 485 | Opex increasing 1% after y. 4 |
| Interest rate | 10% | NPV-discount rate | 10,00% | | Battery: | 180 per KWh | 3 600 | | |
| Depreciation/method. TSE | linear 20 years | Duration of credit | 10 | years, repayment of credit | Lifetime 10 years | | | 36 | OPEX: 1% |
| Depreciation/method. Batteries | linear 10 years | Tariff2 | 0,309 | USD/kWh | CAPEX re-investment: | 120 per KWh | | | = in 1000 USD 2400 |
| O&M cost | 5% of CAPEX for TSE, 2% of Capex for PV, increasing 1% p.a. after year 4 | | | | PV | 770 per MW | 3 465 | | Capacity Factor: 16% |
| Average load (MW): | 2,52 | Annual Generation | 22075 | MWh | OPEX: 2% of CAPEX | | | 34,7 | Opex 2% |
| Peak load (MW) | 4 | | | | Total | | 36 765 | 1555,65 | |

**Pay-back scenario (4,5 MW TSE + 4,5 MW PV+ 20 MWh battery storage)**

| year | 0)* | 1 | 2 | 3 | 4 | 5 | 6 | 7 | 8 | 9 | 10 | 11 | 12 | 13 | 14 | 15 | 16 | 17 | 18 | 19 | 20 |
|---|---|---|---|---|---|---|---|---|---|---|---|---|---|---|---|---|---|---|---|---|---|
| MWh sold | | 22 075 | 22 075 | 22 075 | 22 075 | 22 075 | 22 075 | 22 075 | 22 075 | 22 075 | 22 075 | 22 075 | 22 075 | 22 075 | 22 075 | 22 075 | 22 075 | 22 075 | 22 075 | 22 075 | 22 075 |
| Tariff €/KWh (UCME)** | | 0,309 | 0,309 | 0,309 | 0,309 | 0,309 | 0,309 | 0,309 | 0,309 | 0,309 | 0,309 | 0,309 | 0,309 | 0,309 | 0,309 | 0,309 | 0,309 | 0,309 | 0,309 | 0,309 | 0,309 |
| Sales income (1000 USD) | | 6 821,2 | 6 821,2 | 6 821,2 | 6 821,2 | 6 821,2 | 6 821,2 | 6 821,2 | 6 821,2 | 6 821,2 | 6 821,2 | 6 821,2 | 6 821,2 | 6 821,2 | 6 821,2 | 6 821,2 | 6 821,2 | 6 821,2 | 6 821,2 | 6 821,2 | 6 821,2 |
| - O&M cost (1000 USD) | | -1 555,7 | -1 555,7 | -1 555,7 | -1 555,7 | -1 571,2 | -1 586,9 | -1 602,8 | -1 618,8 | -1 635,0 | -1 651,4 | -1 667,9 | -1 684,5 | -1 701,4 | -1 718,4 | -1 735,6 | -1 752,9 | -1 770,5 | -1 788,2 | -1 806,1 | -1 824,1 |
| - Battery replacement (1000 USD) | | | | | | | | | | | -2 400,0 | | | | | | | | | | |
| - Depreciation (1000 USD) | | -1 838,3 | -1 838,3 | -1 838,3 | -1 838,3 | -1 838,3 | -1 838,3 | -1 838,3 | -1 838,3 | -1 838,3 | -1 838,3 | -1 838,3 | -1 838,3 | -1 838,3 | -1 838,3 | -1 838,3 | -1 838,3 | -1 838,3 | -1 838,3 | -1 838,3 | -1 838,3 |
| EBIT (1000 USD) | | 3 427,3 | 3 427,3 | 3 427,3 | 3 427,3 | 3 411,8 | 3 396,1 | 3 380,2 | 3 364,2 | 3 348,0 | 931,6 | 3 315,1 | 3 298,4 | 3 281,6 | 3 264,6 | 3 247,4 | 3 230,0 | 3 212,5 | 3 194,8 | 3 176,9 | 3158,864831 |
| Loan Annuity (1000 USD) | | -4 188,3 | -4 188,3 | -4 188,3 | -4 188,3 | -4 188,3 | -4 188,3 | -4 188,3 | -4 188,3 | -4 188,3 | -4 188,3 | | | | | | | | | | |
| Depreciation (1000 USD) | | 1 838,3 | 1 838,3 | 1 838,3 | 1 838,3 | 1 838,3 | 1 838,3 | 1 838,3 | 1 838,3 | 1 838,3 | 1 838,3 | 1 838,3 | 1 838,3 | 1 838,3 | 1 838,3 | 1 838,3 | 1 838,3 | 1 838,3 | 1 838,3 | 1 838,3 | 1 838,3 |
| Equity (1000 USD) | 11 030 | | | | | | | | | | | | | | | | | | | | |
| Cash Flow/Result - Investor (1000 USD) | 11 030 | 1 077,3 | 1 077,3 | 1 077,3 | 1 077,3 | 1 061,7 | 1 046,0 | 1 030,1 | 1 014,1 | 997,9 | -1 418,5 | 5 153,4 | 5 136,7 | 5 119,8 | 5 102,8 | 5 085,6 | 5 068,3 | 5 050,8 | 5 033,1 | 5 015,2 | 4 997,1 |
| Cash flow after potential tax | 11 030 | 1 077,3 | 1 077,3 | 1 077,3 | 1 077,3 | 1 061,7 | 941,4 | 927,1 | 912,7 | 898,1 | -1 276,6 | 4 638,0 | 4 623,0 | 4 607,9 | 4 592,5 | 4 577,1 | 4 561,5 | 4 545,7 | 4 529,8 | 4 513,7 | 4 497,4 |
| Cumulated Cash Flow (1000 USD) | 11 030 | -9 952,2 | -8 875,0 | -7 797,7 | -6 720,5 | -5 658,8 | -4 612,8 | -3 582,7 | -2 568,6 | -1 570,7 | -2 989,2 | 2 164,2 | 7 300,9 | 12 420,7 | 17 523,6 | 22 609,2 | 27 677,5 | 32 728,3 | 37 761,3 | 42 776,5 | 47 773,6 |
| Cash Flow 100%equity (1000 USD) | 36 765 | 5 265,6 | 5 265,6 | 5 265,6 | 5 265,6 | 5 250,0 | 5 234,3 | 5 218,4 | 5 202,4 | 5 186,2 | 2 769,9 | 5 153,4 | 5 136,7 | 5 119,8 | 5 102,8 | 5 085,6 | 5 068,3 | 5 050,8 | 5 033,1 | 5 015,2 | 4 997,1 |
| Costs: OPEX+CAPEX (1000 USD) | 36 765 | -1 555,7 | -1 555,7 | -1 555,7 | -1 555,7 | -1 571,2 | -1 586,9 | -1 602,8 | -1 618,8 | -1 635,0 | -4 051,4 | -1 667,9 | -1 684,5 | -1 701,4 | -1 718,4 | -1 735,6 | -1 752,9 | -1 770,5 | -1 788,2 | -1 806,1 | -1 824,1 |

* Year 0: No operation, only investment, loan disbursement and payments end of year

** End user tariff plus UCME subsidy adapted to minimum ROE of 15% (with present tariffs ROE is higher)

| Return on equity: | 15,0% | after tax | 14,2% |
|---|---|---|---|
| NPV on total capital. 1000 USD | 6 570,4 | IRR on total capital | 12,6% |

| | | |
|---|---|---|
| discounted MWh | 187 939 | LCOE -274,0 USD/MWh |
| discounted Opex+Capex 1000 USD | -51 503 | |
| annual saving potential vs. base case (total) | 1 938 146 | USD undiscounted |
| annual savings because of lower tariffs | 3 134 678 | USD undiscounted |

| | Total | Total discounted | |
|---|---|---|---|
| MWH sold | 441 504 | 187 939 | |
| Sales income | 136 425 | 58 073 | 1000 USD |
| Total OPEX +CAPEX | -72 503 | -51 503 | 1000 USD |
| EBIT | -72 503 | 27 685 | 1000 USD |

**Figure A1.** Project results.

## Appendix B

### Socioeconomic calculation

| Year | 0 | 1 | 2 | 3 | 4 | 5 | 6 | 7 | 8 | 9 | 10 | 11 | 12 | 13 | 14 | 15 | 16 | 17 | 18 | 19 | 20 |
|---|---|---|---|---|---|---|---|---|---|---|---|---|---|---|---|---|---|---|---|---|---|
| MWh sold | | 22 075 | 22 075 | 22 075 | 22 075 | 22 075 | 22 075 | 22 075 | 22 075 | 22 075 | 22 075 | 22 075 | 22 075 | 22 075 | 22 075 | 22 075 | 22 075 | 22 075 | 22 075 | 22 075 | 22 075 |
| Average tariff in Philippines minus tax*) | | 0,135 | 0,135 | 0,135 | 0,135 | 0,135 | 0,135 | 0,135 | 0,135 | 0,135 | 0,135 | 0,135 | 0,135 | 0,135 | 0,135 | 0,135 | 0,135 | 0,135 | 0,135 | 0,135 | 0,135 |
| Sales income 1000 USD | | 2 980,2 | 2 980,2 | 2 980,2 | 2 980,2 | 2 980,2 | 2 980,2 | 2 980,2 | 2 980,2 | 2 980,2 | 2 980,2 | 2 980,2 | 2 980,2 | 2 980,2 | 2 980,2 | 2 980,2 | 2 980,2 | 2 980,2 | 2 980,2 | 2 980,2 | 2 980,2 |

**Base Case (Diesel)**

| Year | 0 | 1 | 2 | 3 | 4 | 5 | 6 | 7 | 8 | 9 | 10 | 11 | 12 | 13 | 14 | 15 | 16 | 17 | 18 | 19 | 20 |
|---|---|---|---|---|---|---|---|---|---|---|---|---|---|---|---|---|---|---|---|---|---|
| - re-investment/res.value -VAT (1000 USD | 726,0 | - | - | - | - | 2 068,0 | - | - | - | - | - | - | - | - | - | 2 274,8 | - | - | - | - | 1 016,4 |
| - diesel fuel cost (1000 USD) minus tax**) | | -5110,2 | -5110,2 | -5110,2 | -5110,2 | -5110,2 | -3980,5 | -3980,5 | -3980,5 | -3980,5 | -3980,5 | -3980,5 | -3980,5 | -3980,5 | -3980,5 | -3980,5 | -3980,5 | -3980,5 | -3980,5 | -3980,5 | -3980,5 |
| - Other O&M cost (1000 USD) min VAT | | -103,4 | -103,4 | -103,4 | -103,4 | -103,4 | -103,4 | -103,4 | -103,4 | -103,4 | -103,4 | -103,4 | -103,4 | -103,4 | -103,4 | -103,4 | -103,4 | -103,4 | -103,4 | -103,4 | -103,4 |
| Total Cost (1000 USD) | - 726,0 | - 5 213,6 | - 5 213,6 | - 5 213,6 | - 5 213,6 | - 7 281,6 | - 4 083,9 | - 4 083,9 | - 4 083,9 | - 4 083,9 | - 4 083,9 | - 4 083,9 | - 4 083,9 | - 4 083,9 | - 4 083,9 | - 6 358,7 | - 4 083,9 | - 4 083,9 | - 4 083,9 | - 4 083,9 | - 3 067,5 |
| **Total Cash Flow** | - 726,0 | - 2 233,4 | - 2 233,4 | - 2 233,4 | - 2 233,4 | - 4 301,4 | - 1 103,8 | - 1 103,8 | - 1 103,8 | - 1 103,8 | - 1 103,8 | - 1 103,8 | - 1 103,8 | - 1 103,8 | - 1 103,8 | - 3 378,6 | - 1 103,8 | - 1 103,8 | - 1 103,8 | - 1 103,8 | 87,4 |

| | | | | | discounted MWh | | **187 939** | LCOE | | | **-220,6** | USD/MWh | | | |
| discounted cost | | | | | | | **-41 455** | NPV (1000 USD) | | | **-16 083** | | | | |

| CO2 eq emissions t/MWh at 35% | 0,76467 | Assumed CO2 eq price. USD/Ton | 25,0 |
| CO2 eq emissions t/MWh at 45% | 0,59474 |

| Year | 0 | 1 | 2 | 3 | 4 | 5 | 6 | 7 | 8 | 9 | 10 | 11 | 12 | 13 | 14 | 15 | 16 | 17 | 18 | 19 | 20 |
|---|---|---|---|---|---|---|---|---|---|---|---|---|---|---|---|---|---|---|---|---|---|
| = CO2 eq emissions in tons | | 16880 | 16880 | 16880 | 16880 | 16880 | 13129 | 13129 | 13129 | 13129 | 13129 | 13129 | 13129 | 13129 | 13129 | 13129 | 13129 | 13129 | 13129 | 13129 | 13129 |
| = cost of CO2 eq emissions 1000 USD | | -422,0 | -422,0 | -422,0 | -422,0 | -422,0 | -328,2 | -422,0 | -422,0 | -422,0 | -422,0 | -422,0 | -422,0 | -422,0 | -422,0 | -422,0 | -422,0 | -422,0 | -422,0 | -422,0 | -422,0 |
| = Total cost after CO2 eq emissions | - 726,0 | - 5 635,6 | - 5 635,6 | - 5 635,6 | - 5 635,6 | - 7 703,6 | - 4 412,2 | - 4 505,9 | - 4 505,9 | - 4 505,9 | - 4 505,9 | - 4 505,9 | - 4 505,9 | - 4 505,9 | - 4 505,9 | - 6 780,7 | - 4 505,9 | - 4 505,9 | - 4 505,9 | - 4 505,9 | 3 489,5 |
| = total cash flow after CO2 eq emissions | - 726,0 | - 2 655,4 | - 2 655,4 | - 2 655,4 | - 2 655,4 | - 4 723,4 | - 1 432,0 | - 1 525,8 | - 1 525,8 | - 1 525,8 | - 1 525,8 | - 1 525,8 | - 1 525,8 | - 1 525,8 | - 1 525,8 | - 3 800,6 | - 1 525,8 | - 1 525,8 | - 1 525,8 | - 1 525,8 | 509,4 |

| | | | | discounted MWh | | **187 939** | LCOE | | | **-239,4** | USD/MWh |
| discounted cost | | | | | | **-44 994** | NPV (1000 USD) | | | **-19 623** | |

| | Total | Total discounted | |
|---|---|---|---|
| MWH sold | 441 504 | 187 939 | |
| Sales income | 59 603 | 25 372 | 1000 USD |
| Total OPEX +CAPEX | -91 379 | -41 455 | 1000 USD |
| CO2 emissions | 281 336 | 125 995 | tons |
| CO2 Emission cost | -8 346 | -3 540 | 1000 USD |

### Supply mix: 4,5 MW TSR + 4,5 MW PV +20 MWh battery-storage

| Year | 0 | 1 | 2 | 3 | 4 | 5 | 6 | 7 | 8 | 9 | 10 | 11 | 12 | 13 | 14 | 15 | 16 | 17 | 18 | 19 | 20 |
|---|---|---|---|---|---|---|---|---|---|---|---|---|---|---|---|---|---|---|---|---|---|
| Sales income 1000 USD | 0 | 2980,2 | 2980,2 | 2980,2 | 2980,2 | 2980,2 | 2980,2 | 2980,2 | 2980,2 | 2980,2 | 2980,2 | 2980,2 | 2980,2 | 2980,2 | 2980,2 | 2980,2 | 2980,2 | 2980,2 | 2980,2 | 2980,2 | 2980,2 |
| Capex+ batt.repl. min 25 % VAT+imp.duty | -27573,75 | | | | | | | | | | -1800 | | | | | | | | | | |
| OPEX min. VAT. 1000 US $ | | -1369,0 | -1369,0 | -1369,0 | -1369,0 | -1382,7 | -1396,5 | -1410,5 | -1424,6 | -1438,8 | -1453,2 | -1467,7 | -1482,4 | -1497,2 | -1512,2 | -1527,3 | -1542,6 | -1558,0 | -1573,6 | -1589,3 | -1605,2 |
| Total Cost | -27573,75 | -1369,0 | -1368,972 | -1368,972 | -1368,972 | -1382,66172 | -1396,48834 | -1410,4532 | -1424,5578 | -1438,8033 | -3253,19136 | -1467,7233 | -1482,4005 | -1497,2245 | -1512,1968 | -1527,3187 | -1542,5919 | -1558,0178 | -1573,598 | -1589,334 | -1605,2 |
| **Total cash flow** | -27573,8 | 1611,2 | 1611,2 | 1611,2 | 1611,2 | 1597,5 | 1583,7 | 1569,7 | 1555,6 | 1541,3 | -273,0 | 1512,4 | 1497,8 | 1482,9 | 1468,0 | 1452,8 | 1437,6 | 1422,1 | 1406,6 | 1390,8 | 1374,9 |

| | | | | | discounted MWh | | **187 939** | LCOE | | | **-215,1** | USD/MWh |
| | | | | | discounted cost | | **-40 423** | NPV (1000 USD) | | | **-15 051** | |

*) between 11,7 $-cent for business and 19,1 Cent for households, 12% VAT deducted = 13,552

**) 6 Pesos p.liter minus 12 % VAT

| diesel price net of taxes | 40,612 | Pesos = | 0,81224 | USD/l |

| | Total | Total discounted | |
|---|---|---|---|
| MWH sold | 441 504 | 187 939 | |
| Sales income | 59 603 | 25 372 | 1000 USD |
| Total OPEX +CAPEX | -58 711 | -40 423 | 1000 USD |

**Figure A2.** Socioeconomic calculation.

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
