# Peer review of "Financial and Economic Assessment of Tidal Stream Energy—A Case Study"

_ijfs, doi:10.3390/ijfs8030048_

Round 1

Reviewer 1 Report

A well written work. A very interesting case to read. Also useful for educational use!

Minor comments
- The abstract is a bit too long. A decrease around 300 words would help the reader.
- pag 5 there is a reference error
- page 6ss paragraphs 3.1 and 3.2 refer to NPV and WACC too extensively. The legibility of the work is affected. It also lowers the scientific level almost to a didactic level. If possible, better refer to other sources of literature and focus on the discussion of interesting aspects such as the choice of r and Beta.
-page 10 error reference
-page 11 error reference
-pag 12 "but" in capital letters after the point
-page 13 2 error reference
- paragraph 4.4 financial evaluation is a bit short, also in consideration of the job title. maybe part of the discussion text can be moved to this section?
pag 14 error reference
- in the text there is a lot of use of incidents between - -, reading is not always easy
- the conclusions articulated in points are not pleasing to the eye, but are clear and convincing. I recommend deleting the bulleted list and articulating it with paragraphs.

Author Response

The abstract is a bit too long. A decrease around 300 words would help the reader o.k.reduced to 276 words.
- pag 5 there is a reference error- not clear
- page 6ss paragraphs 3.1 and 3.2 refer to NPV and WACC too extensively. The legibility of the work is affected. It also lowers the scientific level almost to a didactic level. If possible, better refer to other sources of literature and focus on the discussion of interesting aspects such as the choice of r and Beta.o,k.indeed a very helpful hint. I have tried my best to shorten it, however, since the target group is also non economists, however, some explanation was deemed necessary
-page 10 error reference o.k.
-page 11 error reference o.k.
-pag 12 "but" in capital letters after the point o.k
-page 13 2 error reference not clear
- paragraph 4.4 financial evaluation is a bit short, also in consideration of the job title. maybe part of the discussion text can be moved to this section? I think there is a misunderstanding. The chapter is called financing, not financial evaluation. The financial evaluation is above, but make this clear, I have inserted the headline "financial ecaluation" on page 10 
pag 14 error reference
- in the text there is a lot of use of incidents between - -, reading is not always easy
- the conclusions articulated in points are not pleasing to the eye, but are clear and convincing. I recommend deleting the bulleted list and articulating it with paragraphs. o.k.

Reviewer 2 Report

This paper presents exemplary case study for making assessment of tidal stream energy from the financial and economic perspectives. Overall, the manuscript was well-organised and written to read easily. However, the author had better to make emphasis on the originality of this paper and marginal contribution to both academic and practical society. Since the research questions come from the consulting projects and the main goal of this article is to find the best combination of renewable energy technologies to maximize profit from the various cases, it seems that practical insights and contributions would be greater than those of academic aspects. For example, socioeconomic calculation could be emphasized more for potential investors to judge business validity because it can improve the quality of decision making regarding energy business projects.

As a minor point, there are so many punctuation marks(or full stops) which should be replaced with comma(,). For instance, line 295, 308, 314, 330, 335 and 337 etc..  Also, typos like lines 300( project missing), 302(this this) and 361(cost cost) etc.. Except for these, there are a number of parts that should be modified. Please read carefully and have them corrected. Then it could be publishable. 

Author Response

Since the research questions come from the consulting projects and the main goal of this article is to find the best combination of renewable energy technologies to maximize profit from the various cases, it seems that practical insights and contributions would be greater than those of academic aspects.

Actually the applied character of this essay is purposely chosen, since I believe that especially in the area of environmental sciences there are already too many academic papers in full beauty but with no practical use 

For example, socioeconomic calculation could be emphasized more for potential investors to judge business validity because it can improve the quality of decision making regarding energy business projects.

Basically you are right, but the financial soundness of a project is still the first step to find investors. There is, by the way, a rather extensive part on the problem of the inclusion of GHG emissions, which are important to be considered, but nevertheless a real "price" is still missing, especially in Asia, where the location of the project should be. Other socioeconomic issues are -in line with e.g. world bank practice- mentioned verbally but not evaluated on purpose, because it is difficult and it opens up ways to manipulate results. 

As a minor point, there are so many punctuation marks(or full stops) which should be replaced with comma(,). For instance, line 295, 308, 314, 330, 335 and 337 etc..  Also, typos like lines 300( project missing), 302(this this) and 361(cost cost) etc.. Except for these, there are a number of parts that should be modified. Please read carefully and have them corrected

Yes, sorry, there was a little accident. While converting commas to full stops in a numeric table, the search and convert function replaced all commas in the entire chapter. I re-changed them manually, but as I could see, there were still some which have been overlooked (and it must have looked strange for the reader). Sorry, but I have tried my best now to change all the remaining wrong full stops and also to do some editing to the remaining text
